# HiMoRNA: A Comprehensive Database of Human lncRNAs Involved in Genome-Wide Epigenetic Regulation

**DOI:** 10.3390/ncrna8010018

**Published:** 2022-02-08

**Authors:** Evgeny Mazurov, Alexey Sizykh, Yulia A. Medvedeva

**Affiliations:** 1Institute of Bioengineering, Research Center of Biotechnology, Russian Academy of Sciences, 117312 Moscow, Russia; mazurovEV@gmail.com; 2School of Biological and Medical Physics, Moscow Institute of Physics and Technology, Dolgoprudny, 141701 Moscow, Russia; sizykh.ad@phystech.edu

**Keywords:** lncRNA, histone modification, database, X inactivation

## Abstract

Long non-coding RNAs (lncRNAs) play an important role in genome regulation. Specifically, many lncRNAs interact with chromatin, recruit epigenetic complexes and in this way affect large-scale gene expression programs. However, the experimental data about lncRNA-chromatin interactions is still limited. The majority of experimental protocols do not provide any insight into the mechanics of lncRNA-based genome-wide epigenetic regulation. Here we present the HiMoRNA (Histone-Modifying RNA) database, a resource containing correlated lncRNA–epigenetic changes in specific genomic locations genome-wide. HiMoRNA integrates a large amount of multi-omics data to characterize the effects of lncRNA on epigenetic modifications and gene expression. The current release of HiMoRNA includes more than five million associations in humans for ten histone modifications in multiple genomic loci and 4145 lncRNAs. HiMoRNA provides a user-friendly interface to facilitate browsing, searching and retrieving of lncRNAs associated with epigenetic profiles of various chromatin loci. Analysis of the HiMoRNA data suggests that several lncRNA including JPX might be involved not only in regulation of XIST locus but also in direct establishment or maintenance of X-chromosome inactivation. We believe that HiMoRNA is a convenient and valuable resource that can provide valuable biological insights and greatly facilitate functional annotation of lncRNAs.

## 1. Introduction

Transcription in higher organisms is pervasive and produces a wide variety of noncoding RNAs [1], with long non-coding RNAs (lncRNAs) being one of the most populated classes. Classically, lncRNAs are defined as transcripts longer than 200 nt in length lacking protein-coding capacity. LncRNAs are usually lowly expressed, highly tissue-specific and rarely evolutionary conserved leading to difficulties in functional annotation [2,3]. However, lncRNAs are heavily regulated [4,5] which supports their functionality. Indeed, it has been shown that lncRNAs function via surprisingly diverse molecular mechanisms [6,7] and have a vital role in various biological processes [8,9].

Many lncRNAs interact with chromatin and are essential for epigenetic control of specific genome loci as well as organization of the chromosomes [10,11,12,13]. Given that, identification of chromatin-interacting lncRNAs and their genome targets could be a critical first step for dissecting lncRNA function in epigenetic regulation. Many low-throughput experiments aimed at investigating lncRNA–chromatin interactions have been performed over the years [14,15,16]. With advances in next-generation sequencing, several high-throughput techniques (ChIRP-seq [17], CHART-seq [18], MARGI [19], GRID-seq [20], ChAR-seq [21], RADICL-seq [22], Red-C [23], have been employed to detect the binding sites of chromatin-interacting lncRNAs genome-wide. Experimentally validated lncRNA–chromatin interactions that could facilitate the understanding of the functional role of lncRNAs have been integrated in LnChrom [24], suggesting that more than 2300 lncRNAs could interact with the chromatin in at least one cell type. However, high-throughput methods suffer from a high level of false-positives, since sporadic contacts could be also detected in the experiments. These methods also have issues with detecting contacts for lowly expressed lncRNAs. Additionally, high-throughput data on lncRNA-chromatin interactions is limited to a specific cell type and does not uncover a mechanism of lncRNA in regulation of histone modifications.

To address this problem computationally we developed an approach to search for lncRNAs whose presence in the cell is significantly correlated with the presence of epigenetic modification in a specific genome locus. Here we report the HiMoRNA (Histone-Modifying RNA) database, a resource containing coordinated lncRNA–epigenetic changes genome-wide. HiMoRNA integrates a large amount of multi-omics data to characterize the effects of lncRNA on epigenetic modifications and gene expression. The current release of HiMoRNA includes 5,640,250 associations in humans for the ten most studied histone modifications and 4145 lncRNA. HiMoRNA provides a user-friendly interface to facilitate browsing, searching and retrieving of lncRNAs associated with specific chromatin loci. Analysis of the HiMoRNA data suggests that several lncRNA, including JPX, might be involved not only in regulation of XIST locus but also in direct establishment or maintenance of X-chromosome inactivation. We believe that HiMoRNA is a convenient and valuable resource that can provide valuable biological insights and greatly facilitate functional annotation of lncRNAs.

Availability: http://himorna.fbras.ru (accessed on 19 November 2021).

## 2. Materials and Methods

All analyses were performed using human genome build hg38 (https://www.ncbi.nlm.nih.gov/assembly/GCF_000001405.26/, accessed on 19 November 2021) and GENCODE gene annotation version 31 (https://www.gencodegenes.org/human/release_31.html, accessed on 19 November 2021).

### 2.1. Histone Modification Data

We downloaded bam files (ChIP-seq, signals and controls) for ten histone modification marks (H3K27ac, H3K27me3, H3K36me3, H3K4me1, H3K4me2, H3K4me3, H3K9ac, H3K9me3, H3K79me2, H4K20me1) having the highest number of biosamples (Appendix A) from the ENCODE portal [25,26]. We used only those biosamples for which RNA-seq data was also available. Using the NCIS package [27], we estimated a normalization factor for each ChIP-seq experiment and called peaks with MACS2 [28] (FDR-corrected *p*-value < 0.05) for matching pairs of signal-control experiments for the same biosample. ChIP-seq peaks detected in all biosamples for one histone modification were merged using bedtools [29] to standardize the peak boundaries across different cell types. Coordinates of the merged peaks were used as a universal genome annotation for this histone mark and read counts were obtained within each region of the universal annotation for each biosample using bedtools. Read counts were normalized using DESeq2 [30]. We associated peaks to genes from the gencode.v31.annotation if a peak was located within 1000 bp to a gene with ChIPpeakAnno [31] (Appendix A).

### 2.2. Transcriptome Data

We downloaded RNA-seq data (gene counts) from the ENCODE portal for the same biosamples for which ChIP-seq data was used. Genes counts were normalized using DESeq2 separately for each group of RNA-seq samples matching ChIP-seq samples for every histone modification. As a list of lncRNAs we used gencode.v31.long_noncoding_RNAs and normalized the gene expression of each dataset with DESeq2. For each histone mark we further analyzed only a set of lncRNAs expressed in at least ten biosamples.

### 2.3. Correlation Analysis

To select lncRNAs potentially involved in epigenetics regulation of a specific locus across multiple cell types we calculated the Spearman correlation coefficient (SCC) between all lncRNAs expression vectors and all ChIP-Seq peaks over all samples available for every histone modification. Only significant correlations (BH correction, FDR<0.05) for each lncRNA are provided in the database.

Per chromosome distribution of correlated ChIP-seq peaks, for each lncRNA and histone modification in the database we tested if correlated peaks are non-uniformly distributed across different chromosomes by performing Fisher exact test for peaks on one chromosome against all others (BH correction, FDR<0.05).

## 3. Results

### 3.1. Database Statistics

The current release of HiMoRNA contains 5,640,250 significantly correlated lncRNA-genomic loci in humans for ten histone modifications and 4145 lncRNA. In order to characterize the records of lncRNA-genome locus we also gathered a series of metadata for each interaction.

### 3.2. Web Interface

Data browsing, searching and retrieving HiMoRNA provides a user-friendly web interface that allows users to browse, query and download the lncRNA–locus correlation data Figure 1. In order to provide easy access we added a choice of filters for fast browsing and searching. The user can choose histone modification(s), lncRNA name(s), gene names, coordinates of the regions (in bed format), the SCC sign and value allowing users to perform a customized search for the contents of interest. The user is required to select at least a histone modification but to make the search faster we recommend selecting one or only a few RNAs or uploading the genomic regions. The server will return the related lncRNA-loci pairs. The user can also download the whole database content from the home page as well as the results of the custom search from the corresponding page.

The initial output of the browsing or searching contains basic information about each correlated lncRNA-locus and is comprised of the histone modification name, lncRNA name, chromosome, start and end of the correlated loci (ChIP-seq peak), target gene name, and SCC value. HiMoRNA also provides a download button allowing users to retrieve all of the interaction data.

To further explore the interaction of interest, users can click on a histone modification, an lncRNA, a gene or a correlation in the table to launch a new page that contains the corresponding metadata and visualizations. LncRNA page provides information about its expression in all cell types used in HiMoRNA, per chromosome distribution of correlated peaks for each histone modification separately and a summary table of all peaks correlated with this lncRNA. The histone modification page provides information about chromosomal distribution of ChIP-seq peaks and a table of all lncRNA correlated with at least one peak of this modification. The gene page provides information about its expression in all cell types used in HiMoRNA, a table of all histone modification peaks associated with the gene and a summary table of all peaks associated with this gene and at the same time correlated with any lncRNA. The correlation page provides detailed information about how this correlation has been obtained, including lncRNA expression and ChIP-seq peak signal values in all the cell types used to calculate SCC.

### 3.3. Implementation

LncRNA-locus correlations and the results of multi-omics data set analysis were stored and queried by neo4j. The web interface was implemented in JavaScript and Vue and has been tested on Firefox, Chrome and Safari.

Source code and database dumps in different formats are available at https://github.com/lab-medvedeva/himorna-frontend (accessed on 19 November 2021) (platform frontend) and https://github.com/lab-medvedeva/himorna-backend (accessed on 19 November 2021) (platform backend).

## 4. Discussion

HiMoRNA provides a comprehensive collection of lncRNA–genomic loci for which lncRNA expression is significantly correlated with the histone modification signal across multiple cell and tissue types. HiMoRNA currently contains over five million correlations for ten histone modifications and 4145 lncRNA in humans. For each lncRNA, histone modification and gene we summarized the metadata to facilitate the study of lncRNAs function in epigenetic regulation. HiMoRNA also provides a user-friendly web interface for customized searching, browsing and accessing the interaction data.

Deeper investigation of per chromosome distribution of ChIP-seq peaks associated with specific lncRNAs demonstrated significant nonuniformity. For seven out of ten studied histone modifications, at least one lncRNA demonstrated an enrichment of correlated ChIP-seq peaks at one of the chromosomes. The most striking cases include lncRNA JPX and TSIX for which the X-chromosome contains the majority of the correlated peaks for H3K9me3 and H3K27me3, respectively (Figure 2), while the length-normalized number of peaks for each modification is not dramatically different (SF9).

X-chromosome inactivation is an essential epigenetic process which is regulated by several lncRNAs, including XIST, the master regulator of X-inactivation initiation. Other lncRNAs have also been implicated in regulating XIST, including TSIX, JPX, and FTX. Tian et al. [32] proposed that XIST is controlled by two RNA switches: TSIX, which represses XIST on the active X chromosome, and JPX, which activates XIST on the inactive X chromosome. Our pipeline shows a strong link between TSIX and H3K27me3, as well as between JPX and H3K9me3. Both H3K27me3 and H3K9me3 are repressive marks and are known to contribute to different stages of X inactivation. Our observations suggest that TSIX and JPX may also contribute not only to regulation of XIST per se but also to the repression of X-chromosome directly by affecting different histone modifications at different stages of repression.

It has been found recently that Jpx acts as a CTCF release factor and in this way affects a 3D structure of the genome by regulating anchor site usage [33]. CTCF is a DNA-binding protein with multiple roles in chromatin architecture and gene regulation [34]. Apart from being an architectural protein CTCF serves as a transcription factor. We hypothesize that the JPX-induced release of CTCF from promoters or enhancers, where this transcription factor serves as an activator, may contribute to the establishment of the repressed chromatin mark H3K9me3, observed in our study.

Finally, we observed several other lncRNAs significantly correlated with the ChIP-seq peaks of a particular histone modification on the X-chromosome (ST3) for which experimental evidence of participation in X-chromosome inactivation is still required.

All the above suggest a possible mechanistic explanation of the role of several lncRNAs in establishing and maintaining the epigenetic profile of inactive X-chromosome. We realize that correlation of the ChIP-seq peaks and lncRNA expression does not guarantee causal relationship between these two events. Still, we hypothesize that at least some of these lncRNA could participate in epigenetic complex formation or attract them to specific X-chromosome loci. This example shows nicely how HiMoRNA enables researchers to translate associations between lncRNA and histone modifications into novel biologic insights.

## 5. Future Development

The area of functional lncRNA annotation is moving fast. We are fully committed to the maintenance and update of HiMoRNA, and we will monitor new data on the role of lncRNA in epigenetic regulation and expand the database as data becomes available. Specifically, we will add experimental data on lncRNA-chromatin interactions available databases to support our predicted lncRNA-genomic loci interactions. We also will incorporate data on lncRNA knockdown effects of gene expression from FANTOM6 [35]. As more experimentally validated data become available, we hope to build several predictive models for lncRNA–histone modification. Furthermore, we will continue to improve the performance of our computer servers for storing and analysing the newly generated data. We expect that these continuous efforts and our commitment to develop and improve HiMoRNA will contribute to the understanding of lncRNA-mediated chromatin regulation.

## Figures and Tables

**Figure 1 ncrna-08-00018-f001:**
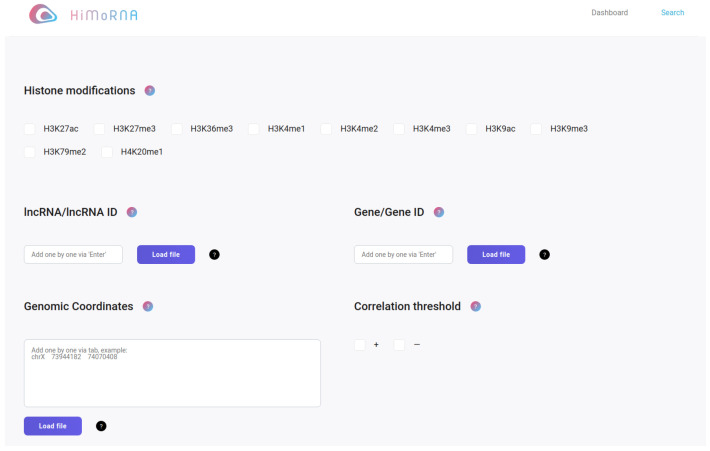
HiMoRNA web-interface. Users can query the resource through selecting lncRNA, gene, genomic regions and setting a threshold for the sign and an absolute value of an SCC.

**Figure 2 ncrna-08-00018-f002:**
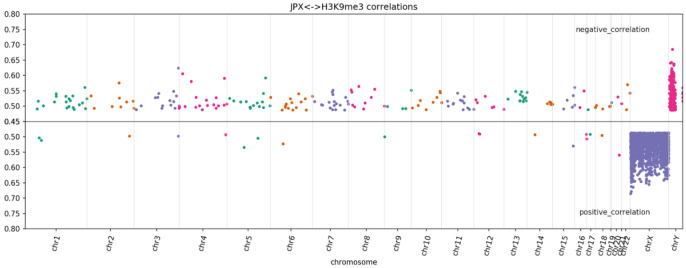
Per chromosomal distribution of lncRNA-correlated peaks of H3K9me3 for JPX.

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
