# Peer review of "HiMoRNA: A Comprehensive Database of Human lncRNAs Involved in Genome-Wide Epigenetic Regulation"

_ncrna, 2022, doi:10.3390/ncrna8010018_

Round 1

Reviewer 1 Report

The work done by the authors is good. These types of databases are required for understanding the functional role of lncRNAs. However, there are some minor mistakes that need to be corrected before the acceptance of the manuscript.

Line 133: Figures, Tables and Schemes: Where is the text for the subheading?

Author Response

We are grateful to the reviewer for the high assessment of our database. We believe that it really can help other researchers. 

R1:
Line 133: Figures, Tables and Schemes: Where is the text for the subheading?
Response:
DONE. Since Figures are provided in the main text and supplementary materials we believe that the subheading should be removed.  

We also performed English spell check as suggested.

Reviewer 2 Report

Mazurov at al present a web accessible database for human lncRNAs and their associations with a selection of the most commonly studied histone marks.

This has the potential to be an extremely valuable tool, however the website wasn't functioning at the time of me testing it. Without being able to run searched for lncRNAs or genes of interest I can't recommend this for publication. This is very disappointing, as I was looking forward to testing the tools.

Major concerns

Web site not functioning - searches don't work

Minor points

No open source code provided for the methods used to generate the database and the website

No plain text download available for the underlying database. This limits the usefulness of this valuable work if not available to researchers so the work can be extended and large scale / programatic searches performed.

A very recent paper in Cell (Oh et al, 2021) should be cited in a revised paper - it puts forward a mechanism for Jpx which is very relevant to this paper.

Author Response

We are very grateful to the reviewer for the appreciation of our work and valuable comments.

R2:
This has the potential to be an extremely valuable tool, however the website wasn't functioning at the time of me testing it. Without being able to run searched for lncRNAs or genes of interest I can't recommend this for publication. This is very disappointing, as I was looking forward to testing the tools.

Response: We are very sorry that the server was down. Now it is up and running. 

Major concerns

Web site not functioning - searches don't work

Response: Now the server is up and running. Please have in mind that since the data is rather big, to run a search one has to specify a mark, an lncRNA (or a list of them), and target region(s).

Minor points

R2:

No open source code provided for the methods used to generate the database and the website

Response:

In response to your request, we published our platform source code in two separated repositories: https://github.com/lab-medvedeva/himorna-frontend (platform frontend) and https://github.com/lab-medvedeva/himorna-backend (platform backend). Our frontend is written mainly with Vue.js framework, which was chosen to speed up the development, and for the backend part we chose Flask Python microframework for server speed and easy further improvements.

R2:
No plain text download available for the underlying database. This limits the usefulness of this valuable work if not available to researchers so the work can be extended and large scale / programatic searches performed.

Response:

The original data size used for database generation is rather huge. On our side, we store them in binary files. It doesn't seem practical to both provide text files for downloading and to use this data for direct search without any database engine. We suggest using a database dump in cypher. The file is humanly readable but at the same time, it can be easily imported into neo4j. The current version of a website has a download button on the main page that allows downloading of the dump. The dump is compressed to speed up downloading and uploading it to the local database.

R2:

A very recent paper in Cell (Oh et al, 2021) should be cited in a revised paper - it puts forward a mechanism for Jpx which is very relevant to this paper.

Response:

We are very grateful to the reviewer for bringing this important work to our attention. We added the following text to the discussion.

It has been found recently that Jpx acts as a CTCF release factor and in this way
affects a 3D structure of the genome by regulating anchor site usage [33]. CTCF is a DNA-binding protein with multiple roles in chromatin architecture and gene regulation [34]. Apart from being an architectural protein CTCF serves as a transcription factor. We hypothesize that the JPX-induced release of CTCF from promoters or enhancers, where this transcription factor serves as an activator, may contribute to the establishment of the repressed chromatin mark H3K9me3, observed in our study.

We also performed minor rearrangements in the results section and spell-checked the paper.

Round 2

Reviewer 2 Report

I was now able to use and test the online version of the HiMoRNA resource. Overall this is a very useful tool and the authors should be commended for providing the database dump and GitHub code. This will increase the usefulness of this resource for researchers wanting to delve further into the data. I, myself found some interesting gene:lncRNA correlations, and access to the raw data enables a deeper dive into the data. There are some minor updates that would make the resource easier and more intuitive to use:

Minor Comments:

ZST is an unusual choice for providing the data, some help text should be provided on how to decompress the data and input into cypher (as suggested)

Rather than the full data download, selections of data would be more useful. For example the data tables per gene or lncRNA.

ln109: dounload -> download

Online plots do not have units for the expression - very important if the data is to be used in other studies

Online tables for specific gene correlations could be improved by providing a download as plain text option. This would be a small amount of data for each gene or lncRNA.

The mouse-over on the online scatter plots should include the names of the gene and lncRNA, not just the raw expression values. This would make exploring the data much easier.

Author Response

I was now able to use and test the online version of the HiMoRNA resource. Overall this is a very useful tool and the authors should be commended for providing the database dump and GitHub code. This will increase the usefulness of this resource for researchers wanting to delve further into the data. I, myself found some interesting gene:lncRNA correlations, and access to the raw data enables a deeper dive into the data.

Response:

We are happy that the reviewer finds our resource valuable and that it helps him to find new insides for the research. To facilitate the use of the database we provided a csv version of a database and added the download button to the main page.

There are some minor updates that would make the resource easier and more intuitive to use:

Minor Comments:

ZST is an unusual choice for providing the data, some help text should be provided on how to decompress the data and input into cypher (as suggested)

Response:

To avoid any complications for the users we moved a cypher dump of the database to project GitHub and provided a plain text (csv) dump for download from the main page of the HiMoRNA website.  

Rather than the full data download, selections of data would be more useful. For example the data tables per gene or lncRNA. 

Response: We provided a download button for the results of a specific search as suggested.

We also added the following phrase to the text:

The user can also download the whole database content from the home page as well as the results of the custom search from the corresponding page.

ln109: dounload -> download

Response: Done

Online plots do not have units for the expression - very important if the data is to be used in other studies

Response:
We added labels for the plots and included units for the expression.

Online tables for specific gene correlations could be improved by providing a download as plain text option. This would be a small amount of data for each gene or lncRNA.

Response:

We provided a download button for the majority of data and search pages to facilitate further analysis. 

The mouse-over on the online scatter plots should include the names of the gene and lncRNA, not just the raw expression values. This would make exploring the data much easier.

Response:

The only scatter plot we have represents a particular link between one lncRNA and a specific histone modification peak. Not all the peaks are linked to genes. But even for a gene-associated peak on one scatter plot we represent values of expression/peak signal for one item. Different dots represent various tissues, not genes. Since the name of the tissue could be rather long we decided to provide these names in a table below the scatter plot, not in the mouse-over.